# Development of Enriched Oil with Polyphenols Extracted from Olive Mill Wastewater

**DOI:** 10.3390/foods12030497

**Published:** 2023-01-21

**Authors:** Vassilis Athanasiadis, Andreas Voulgaris, Konstantinos Katsoulis, Stavros I. Lalas, Ioannis G. Roussis, Olga Gortzi

**Affiliations:** 1Department of Food Science and Nutrition, School of Agricultural Sciences, University of Thessaly, GR-43100 Karditsa, Greece; 2Department of Agriculture Crop Production and Rural Environment, School of Agricultural Sciences, University of Thessaly, GR-38446 Volos, Greece; 3Department of Animal Husbandry and Nutrition, Faculty of Veterinary Medicine, School of Health Sciences, University of Thessaly, GR-43100 Karditsa, Greece; 4Laboratory of Food Chemistry, Department of Chemistry, University of Ioannina, GR-45110 Ioannina, Greece

**Keywords:** antioxidants, cloud point extraction, food industry, enrichment, olive mill wastewater, olive oil, polyphenols, recovery

## Abstract

The extraction of olive oil produces significant residual olive-mill wastewater (OMW). The composition of OMW varies according to the type of olive, the fruit’s ripeness level, and the extraction method (traditional one-pressing system or continuous systems based on centrifugation of the olive pulp). In olive-producing countries, OMW is important for the environment and the economy and is also a low-cost source of polyphenolic compounds with high antioxidant properties. Olive oil’s properties, such as its anti-atherogenic, anti-inflammatory, anti-aging, and immunological modulator effects, have been attributed to its polyphenols. In this study, the cloud point extraction (CPE) method was used to recover polyphenolic compounds from OMW. The total micellar phase of the three recoveries was 24.2% and the total polyphenols (after sonication) was 504 mg GAE/Kg. Furthermore, the addition of polyphenols recovered from OMW enhanced the olive oil and extended its shelf life without changing its organoleptic properties. There was a 42.2% change in polyphenols after 0.5% enrichment of micellar dispersions. Thus, it is suggested that the CPE method could lead to better waste management in the olive oil industry and improve the nutritional quality of food products.

## 1. Introduction

The demand for olive oil is continuously increasing worldwide. In the Mediterranean area, which is the biggest olive-producing area worldwide, a huge amount of agro-industrial wastewater is generated from the olive processing industry [1,2]. Olive mill wastewater (OMW) residues are composed of solid waste—consisting of olive pulp—as well as liquid waste—consisting of vegetables and additional water generated during decantation. OMW has a dark-brown color (that can become black) and a distinctive, potent aroma reminiscent of olives [3]. The olive mill by-products have environmental and economic significance in olive-producing countries. OMW creates high levels of pollutants because of its high organic load and a high content of phytotoxic and antibacterial polyphenolic substances, which resist biological degradation [4]. The treatment and disposal of OMW is becoming a serious environmental problem. Thus, olive oil-producing countries have been facing a serious challenge to find an environmentally friendly and economically viable solution to the handling and disposal of OMW.

An innovative and environmentally friendly technology for extracting bioactive substances, particularly from food, is cloud point extraction (CPE). It is a technique where chemical or biological components are extracted using non-ionic surfactants that, when heated to (or above) a critical temperature, tend to separate from the bulk solution and create clouds [5]. Surfactants are specifically employed as extractants during the CPE process. In the right circumstances, the extraction takes place at a temperature below the cloud point—when the surfactant becomes cloudy (i.e., less soluble than the initial sample or even insoluble)—leading to the separation of two phases: the aqueous phase and the surfactant-rich phase. Due to clouding processes, the technique leads to a preconcentration of analytes (such as polyphenols) in the surfactant-rich phase [6]. Furthermore, it has been demonstrated that the compounds solubilized by the micelle were protected from oxidation [7].

Polyphenolic compounds, sugars, and organic acids make up the majority of OMW. OMW also has rich resources such as potassium and other mineral nutrients, which may be utilized again as fertilizer [8]. However, OMW could be a natural source of antioxidants due to its high content of polyphenolic compounds. Olive oil polyphenols have potentially beneficial nutritional properties for their antioxidant and biological activities such as anti-allergic, anti-inflammatory, anti-cancer, and anti-hypertension [9,10]. The low redox potential of polyphenols, which enables them to act as reducing agents by donating hydrogen or electrons and scavenging free radicals, is the basis of their antioxidant activity [7].

Oil producers aim to produce food products that keep their nutritional value and shelf life over an extended length of time. These factors render antioxidant supplementation a common technique for lowering lipid oxidation in food production [11]. To prevent oxidation in foods containing fats and oils, synthetic antioxidants including butylated hydroxytoluene (BHT), tert-butyl hydroquinone (TBHQ), and butylated hydroxyanisole (BHA) have been employed as food additives. However, it was revealed that these chemical antioxidants have been linked to a variety of health hazards, including cancer and carcinogenesis [12].

Recent research has shown that natural antioxidants can prevent edible oil oxidation [11,12,13]. The enrichment of olive oil with polyphenolic substances can increase the shelf life of olive oil without affecting its organoleptic characteristics. However, the antioxidant/pro-oxidant balance depends on many factors, particularly dose (e.g., concentration) and timing (e.g., storage duration) and it is a very delicate process that can readily be changed [14]. Consequently, the recovery of polyphenolic compounds not only provides an economic opportunity but also lowers the environmental charge of wastewater.

The aims of this study were first to recover the polyphenolic substances from OMW with the application of the CPE method, and secondly to enrich olive oil with the recovered polyphenolic substances. The stability of the polyphenols is an important parameter in this enrichment. Emulsifiers, such as lecithin, were studied for this purpose.

## 2. Materials and Methods

### 2.1. Materials

In this investigation, a sample of olive mill wastewater (*Olea europaea* var. Koroneiki) from the Preveza region (Epirus, Greece) was employed. The OMW sample was specifically collected by the olive oil industry, frozen, and delivered to our lab the same day. Olive oil from a local market in Karditsa (Thessaly, Greece) was used as the matrix for the control sample, composed of virgin and refined olive oils.

### 2.2. Reagents

The 2,2-Diphenyl-1-picrylhydrazyl (DPPH^•^) and α-tocopherol were purchased from Alfa Aesar (Karlsruhe, Germany). Sodium chloride, citric acid, lecithin of soya (>97%), diethyl ether, and isooctane were obtained from Carlo Erba (Milano, Italy). Sodium carbonate anhydrous and sodium hydroxide were from Penta (Prague, Czech Republic). Absolute ethanol, methanol, Folin–Ciocalteu reagent, and gallic acid monohydrate were from Panreac (Barcelona, Spain). Cyclohexane and *n*-hexane were from Sigma-Aldrich (Steinheim, Germany). The deionized water used in the experiments was produced using a deionizing column.

### 2.3. Extraction of Polyphenols from OMW

A method called cloud point extraction (CPE) was used. The CPE method is based on the notion that when a surfactant’s concentration in a solution exceeds that of the critical micellar dispersion, it can generate micellar dispersions [6]. Figure 1 shows the experimental steps of the CPE method. First, the sample had to be defrosted before the solids could be separated using a centrifuge (Digicen 20-R, Orto Alresa, Madrid, Spain) for 10 min at 4500 rpm. After collecting the supernatant, citric acid (50%) was added to bring the pH level to 3.5. The entire sample was divided into three portions, with lecithin as a surfactant added to each of them in concentrations of 1%, 3%, and 5%, respectively, along with 30% salt. An intense magnetic stirrer was used for agitation. After 20 min, the solution was centrifuged for 5 min at 4500 rpm. The upper micellar (surfactant) phase (SP) and the lower aqueous (water) phase (WP) were examined separately. The first recovery of the polyphenolic dispersions occurred at this point. With the water phase containing the non-extracted analytes, the CPE method was performed two more times under the same conditions (2nd and 3rd recovery).

### 2.4. CPE Method Performance Determination

#### 2.4.1. Total Polyphenol Content

According to Chatzilazarou et al. [15], the Folin–Ciocalteu method was used to determine the total polyphenol content (TPC) photometrically. Briefly, 0.5 g of micellar dispersions and 0.5 mL of Folin–Ciocalteu reagent were added to a 25-mL volumetric flask. After 3 min, 1 mL of Na_2_CO_3_ (35%, *w*/*w*) was added. Deionized water was used to fill the flasks to the necessary volume level, and they were then kept in the dark for 60 min. A Shimadzu UV-1700 UV/Vis spectrophotometer (Kyoto, Japan) was used to measure absorption at 750 nm. Standard solutions (1–10 mg/L) were used to create a standard gallic acid curve. TPC was expressed as mg gallic acid equivalents (GAE) per Kg of micellar dispersions.

#### 2.4.2. Radical Scavenging Activity

The DPPH method of Katsoyannos et al. [16], with some modifications, was used to evaluate the radical scavenging activity of the polyphenols extracted in the surfactant phase as well as those still present in the sample following the CPE treatment (water phase). In short, a 1 mL, 0.1 mM solution of methanolic DPPH was added to 4 mL of the sample solution (500, 1000, and 2000 mg/L). The mixture was thoroughly shaken before being let to stand at room temperature in the dark for 30 min while the absorbance was measured at 517 nm. The formula used to determine the % inhibition was: % Inhibition = 100 × (*A*_control_ − *A*_sample_)/*A*_control_, where *A*_control_ and *A*_sample_ are the absorbances. From the scavenging activities (%) vs concentrations of the corresponding sample curve, the IC_50_ was determined.

### 2.5. Enrichment of Olive Oil with Micellar Dispersions

Using an ultrasonic probe (Misonix Sonicator S3000, Qsonica, LLC, Newtown, CT, USA) and a sonication protocol (75% power energy for 60, 75, and 90 min with 0.5 pulse cycles), micellar size reduction was investigated. The CPE method was used to recover micellar dispersions, which were then added to the olive oil to improve it. For enrichment, concentrations of 0.5, 1, and 5% were employed.

### 2.6. Enriched Olive Oil Quality Analysis

#### 2.6.1. Acidity Value

According to Commission Regulation (EEC) No. 2568/91 [17], the method for determining free fatty acids (FFAs) in olive oil was developed.

#### 2.6.2. Refractive Index

A refractometer (Quartz/Digital Abbe refractometer, Medline Scientific Limited, Oxon, UK) was used to measure the sample’s refractive index. The instrument was calibrated with distilled water, which had a refractive index of 1.3333 at 20 °C. 

#### 2.6.3. Colorimetry

The colorimeter was used to determine the color of the samples of olive oil (Lovibond CAM-System 500, The Tintometer Ltd., Amesbury, UK). In order to determine CIELAB color, a sample (25 mL) was placed in a 50 mL beaker and placed in the colorimeter. A psychometric index of lightness, *L**, and chrominance coordinates, *a** and *b**, were defined. Additionally, the formula for calculating chroma, or *C**, was as follows: *C** = [(*a**)^2^ + (*b**)^2^]^½^.

#### 2.6.4. Spectrophotometric Investigation in the Ultraviolet

Utilizing a spectrophotometer, the method of ultraviolet spectrophotometric investigation was carried out following the Commission Implementing Regulation No 299/2013 [18].

#### 2.6.5. Rancimat Method

According to Lalas et al. [19], the Rancimat method was used to test the oxidation stability of the olive oil samples. More specifically, the reaction vessels of the Rancimat 743 (Metrhom LTD, Herisau, Switzerland) were filled with around 3 g of each olive oil. The chosen parameters were a 90 °C temperature and a 15 L/h airflow. The point when the rate of oxidation changes quickly and produces a strong inflection point on the oxidation curve is known as the induction time, and it is represented by the value of the olive oil stability index. This was the induction period (IP), which was provided in hours. Additionally, this formula was used to compute the protection factor (PF): PF = (IP with antioxidant)/(IP without antioxidant). Inhibition of lipid oxidation is indicated by a protection factor larger than one, and the antioxidant activity is improved by a higher PF value.

#### 2.6.6. Differential Scanning Calorimetry

Differential Scanning Calorimetry (DSC) measures the temperatures and heat flows associated with transitions in materials as a function of time and temperature. These measures include both quantitative and qualitative data about physical and chemical changes involving endothermic or exothermic processes, as well as alterations in heat capacity. DSC was used in this investigation to assess the oil’s oxidative stability [20]. A special aluminum pan was used to weigh 5 mg ± 0.5 mg of oil. A blank pan (reference) was added as well. Oxygen flow was 20 mL/min, and oxidative conditions were programmed at Diamond DSC (PerkinElmer Inc., Shelton, CT, USA). The sample and reference were placed on the DSC, and the thermogram was initiated at 40 °C for 1 min, followed by 300 °C at a rate of 10 °C/min. The onset temperature of the oxidation peak (*T*_max_) determines the starting temperature of oxidation.

#### 2.6.7. Extraction of Polyphenolic Compounds from the Enriched Olive Oils

The extraction of polyphenols from the olive oils was performed as described by Kalantzakis et al. [21]. Olive oil samples (1 g) were dissolved in 2 mL of *n*-hexane and 2 mL of a 60:40 (*v*/*v*) methanol/water solution. The final solution was centrifuged at 4500 rpm for 5 min after vigorous agitation. The soluble in methanol/water (polar) fraction of olive oil samples was obtained and used as it was.

#### 2.6.8. Total Polyphenol Content

As previously mentioned, the total polyphenol content (TPC) was determined using the Folin–Ciocalteu method. TPC was expressed as mg gallic acid equivalents (GAE) per Kg of olive oil.

#### 2.6.9. Radical-Scavenging Activity

The DPPH method, as previously mentioned, was used to evaluate the radical-scavenging activity of the polyphenols extracted from the olive oil samples. The capacity to scavenge the DPPH radical was expressed as % inhibition, and the IC_50_ was calculated from the scavenging activities (%) vs. concentrations of the corresponding sample curve.

### 2.7. Statistical Analysis

The results are given for the mean and standard deviation (SD) of three contemporaneous assays. Using IBM SPSS Statistics (Version 26) statistical software package (SPSS Inc., Chicago, IL, USA), a one-way ANOVA was performed to determine whether there were any statistically significant differences between the mean values. A significant level of *p* < 0.05 was utilized.

## 3. Results and Discussion

### 3.1. Recovery of Polyphenols

A lecithin concentration aqueous phase was performed, where the same procedure was employed to recover polyphenols. Due to the low lecithin concentration of 1%, no surfactant micelle was produced. On the other hand, when there was a high concentration of 5% lecithin, this reduced their solubility in water and resulted in the formation of two phases. Both the equilibration temperature (45 °C) and the salt content (30%) were constant (third recovery). The total micellar phase of the three recoveries is expressed using the Total SP column (Figure 2).

Additionally, sonication procedures (75% power energy for 60, 75, and 90 min with a pulse cycle of 0.5) with an ultrasonic probe were used to examine micelle size reduction. Finally, 0.5% of the polyphenol dispersions recovered by the CPE method was added to the oil to enhance it. This result can be explained by Víctor-Ortega et al. [22], who found that the solubilization of phenolic compounds on surfactant micelles became saturated at higher surfactant concentrations, and the polyphenols retention decreased. In addition, according to Sliwa and Sliwa [23], when the surfactant concentration is below the critical micellar concentration (CMC), the solubility of the active substance is low.

In the Katsoyannos et al. [24] study, when 6% of Triton X-114 surfactant was applied, the CPE technique in OMW (after the removal of fatty compounds) produced recoveries higher than 60%. In a different study, Gortzi et al. [25], showed that utilizing simple and subsequent CPE, the total phenol recovery from OMW with various Genapol X-080 concentrations (2, 5, and 20%, *v*/*v*) reached up to 89.5%. Additionally, Katsoyannos et al. [16], used a double-step CPE with 5% + 5% of Tween 80 to recover up to 94.4% of the total phenols from the OMW. Finally, in the Alibade et al. [6] study, lecithin was used as a surfactant while CPE was used to recover phenolic compounds from wine sludge waste (WSL). The surfactant-rich phases showed strong antioxidant activity. Their findings show that CPE and lecithin can be utilized to successfully separate polyphenols from WSL.

### 3.2. Enrichment of Olive Oil with Polyphenols from Olive Oil Wastewater

An alteration in the organoleptic properties of the micellar dispersions was observed after 0.5% *w/v* enrichment (turbidity). This modification led to the final selection of 3% lecithin enrichment. In order to obtain better outcomes, the first recovery was combined with the second and third ones. Additionally, sonication proved successful in reducing micellar dispersions. The oil system responded negatively to amounts of 1% and 5%. After a few hours of enrichment, the precipitate was seen in the oil samples.

The initial sample was subjected to the CPE method, which caused the polyphenol concentration to drop from 3448 mg to 346 mg GAE/Kg (Figure 3). At the molecular level, the reduction in the number of hydrogen bonds between polyphenols and water allowed the incorporation of polyphenols into the micelle at the expense of its solubilization [26]. The micelles were then treated with sonication and methanol to extract the polyphenols, increasing the polyphenol content to 1562 mg GAE/Kg. The final concentration of polyphenols was 504 mg GAE/Kg (after sonication), which demonstrates that a decrease in micellar dispersion results in a drop in the concentration of total polyphenols [27]. Additionally, prolonged ultrasonic time causes particles to dissolve in the solvent, reducing the TPC value [28].

The enrichment of the olive oil sample with the dispersion was then investigated. There was a 42.2% increase in polyphenols (from 60.2 ± 5.6 to 104.1 ± 8.3 mg GAE/Kg after 0.5% enrichment of micellar dispersions). The enriched olive oil is shown in Figure 4, which also shows how the organoleptic characteristics of the olive oil changed. Although a pleasant, fruity odor (aroma) was developed, turbidity was observed without the deposition of sediment.

Samples of micellar dispersions showed a decrease in free radicals after sonication with increasing concentration (Table 1) by the DPPH method. The highest reductions were observed at the highest concentrations for all samples while the 75-son sample displayed the highest value. The 60-son sample shows lower toxicity than the rest of the samples, while the 90-son sample displays the highest. Statistical analysis, however, showed that there was no significant difference (*p* > 0.05) between the three sonication duration levels. It should also be noted that the 60-son and 75-son samples show similar mean inhibitory concentration patterns. In general, the repulsive interactions between the ionic head groups of the surfactant molecules were reduced when salts were added to the surfactant solution. As a result, it was encouraged to form micelles, which might have had an impact on DPPH free-radical scavenging activity [29]. The hydroxytyrosol and tyrosol phenolic compounds are the most prevalent ones in the OMW extracts. According to Karadag et al. [30], OMW’s high hydroxytyrosol content is a contributing factor to its strong antioxidant activities. In their investigation on the recovery of phenolic compounds from OMW, Yangui and Abderrabba [31] concluded that the recovered polyphenols showed strong antioxidant activity and rapid DPPH free-radical scavenging activity.

### 3.3. Quality Control of Enriched Olive Oils

Olive oil, including both refined and virgin oil, should have an acidity of less than 1%, as specified by Commission Implementing Regulation No 299/2013 [18] (Appendix—Characteristics of Olive Oil). All samples fell within the specified range, as seen in Table 2. At 20 °C, the refractive index should range from 1.4677 to 1.4705. The values of *L** and *a** factors did not change noticeably. After the 75- and 90-son samples were included, factor *b** was altered. The specific extinction coefficients were constant with the control olive oil and with the remaining enrichment sample olive oil. The samples of enhanced olive oil had protective factors that were less than a mean value of 1. Each sample demonstrated pro-oxidant activity. However, a 60-son oil sample with an approximate value of 1 was shown to have the highest oxidant efficiency (0.96 ± 0.05).

The exothermic peaks of the extracts used in this investigation were measured using DSC (Figure 5). Thermographic curves that reveal the temperature of the extrapolated commencement of the thermo-oxidation process can be used by DSC to derive oxidation kinetic parameters. The highest oxidation peak on the thermographic curve is *T*_max_. As the sample demonstrates a stronger resistance, the higher the *T*_max_ value. The 60-son oil sample showed the most important antioxidant efficiency.

Due to the preservation of the initial sample (control), a monthly decrease in the total polyphenol content is observed. The initial sample was used for the monthly sampling (Figure 6).

After enrichment, the total polyphenol content increased significantly. The 75-son oil had a significant consistency in polyphenol content, whose growth percentage peaked after four months (Figure 7). The 60-son oil sample was still largely stable after four months, but the 90-son oil sample exhibited significant changes throughout the same period.

After sonication, the number of free radicals in micellar dispersion-enhanced samples decreased when the concentration increased (Table 3). The 75-son oil sample exhibited the greatest reduction in free radicals compared to other samples. The toxicity of the 75-son sample was lower than that of the other samples. Compared to other samples, the 90-son oil sample exhibited the highest level of toxicity, although the 60-son oil and control samples behaved similarly.

Due to its textural and organoleptic properties, negative environmental effects, and management and disposal issues, olive oil wastewater has attracted attention [32]. High amounts of polyphenolic compounds and the organic load of olive oil waste could be the cause of phytotoxicity and changes in soil microbiota [33]. The addition of polyphenols from oil waste to several food matrices has increased both their antioxidant properties and sensory characteristics, although it presents disadvantages as a fertilizer and feed additive. Previous research has shown that a significant quantity of polyphenols remains in the by-products of olive oil production [34,35,36,37]. To optimize the reintroduction of polyphenolic compounds into the food chain, increase their value, and improve the waste management of the olive oil industry, the effective recovery of polyphenolic compounds has been the subject of substantial research [38].

## 4. Conclusions

In our study, the cloud point extraction method, which used lecithin as an emulsifier at a concentration of 3%, produced significant recovery efficiency of micellar dispersions. Micellar dispersion sample sizes decreased following sonication as the concentration increased. The concentration of total polyphenols in olive oil samples rose to 42.2% with the addition of 0.5% micellar dispersions. The 75-son oil sample initially showed stability, but the total polyphenol concentration increased significantly after four months. In addition, a significant reduction in free radicals was observed in this sample compared to other samples. The oil from the 60-son sample demonstrated less toxicity than the other samples in terms of the mean inhibitory concentration in the free radicals. No specific organoleptic characteristics have been noted. The colors of the samples remained unchanged, no sediment was visible, and the aroma of the olive oil was fruity and pleasant. More research is required to optimize the extraction conditions of polyphenolic components from olive mill wastewater. Their use could lead to better waste management in the olive oil industry as well as improvements in the nutritional quality of food products.

## Figures and Tables

**Figure 1 foods-12-00497-f001:**
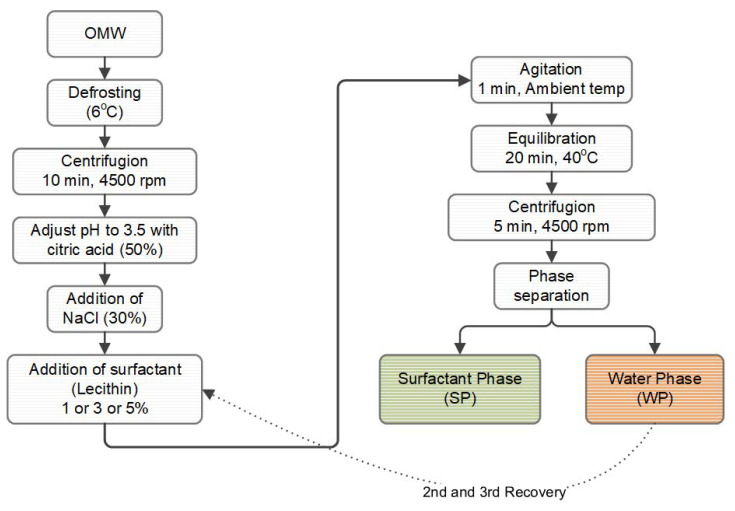
Experimental steps of the CPE method for the extraction of polyphenols from OMW.

**Figure 2 foods-12-00497-f002:**
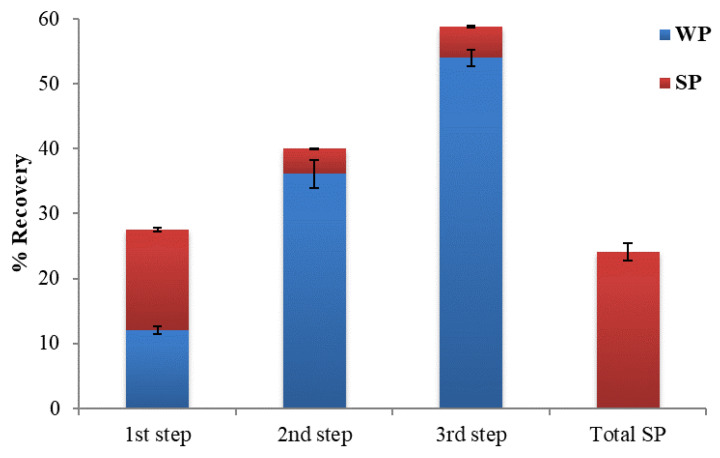
Recovery (%) of polyphenols from OMW with 3% lecithin; WP stands for the Water Phase, SP for the Surfactant Phase, and Total SP for the total micellar phase recovery; standard deviations are presented with error bars.

**Figure 3 foods-12-00497-f003:**
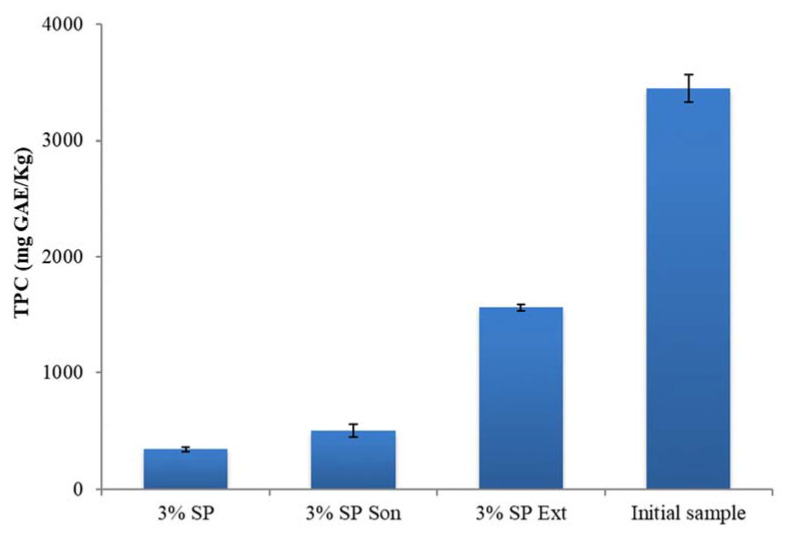
Changes in the concentration of total polyphenols; SP for the Surfactant Phase, and Son for the sonication treatment; Standard deviations are presented with error bars.

**Figure 4 foods-12-00497-f004:**
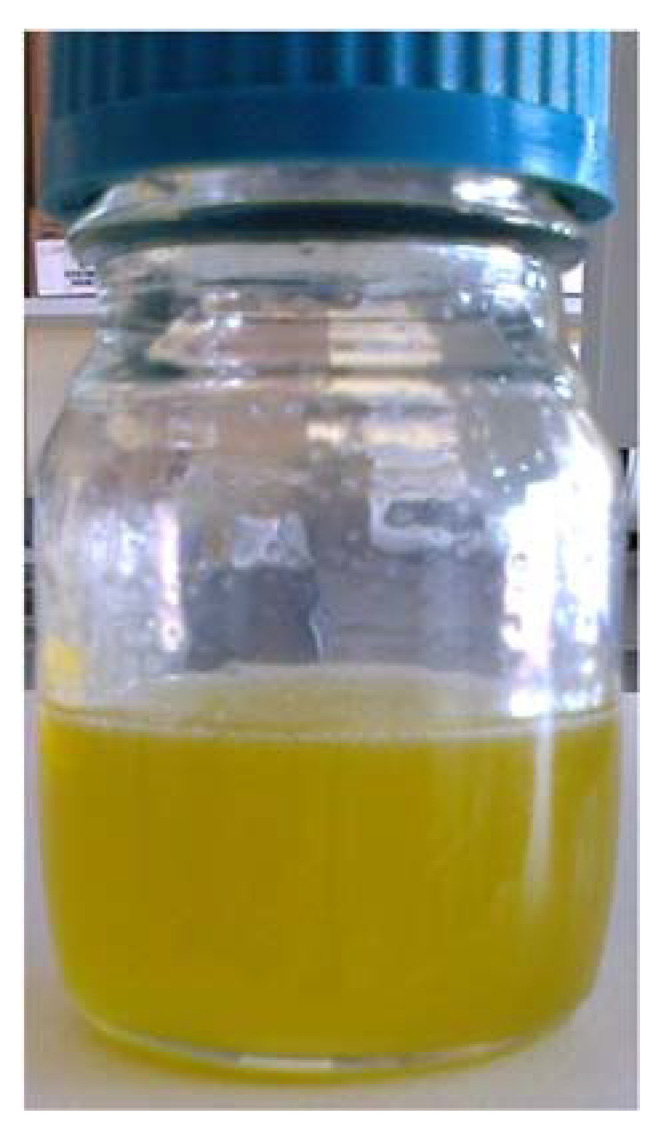
The enriched olive oil with OMW polyphenols.

**Figure 5 foods-12-00497-f005:**
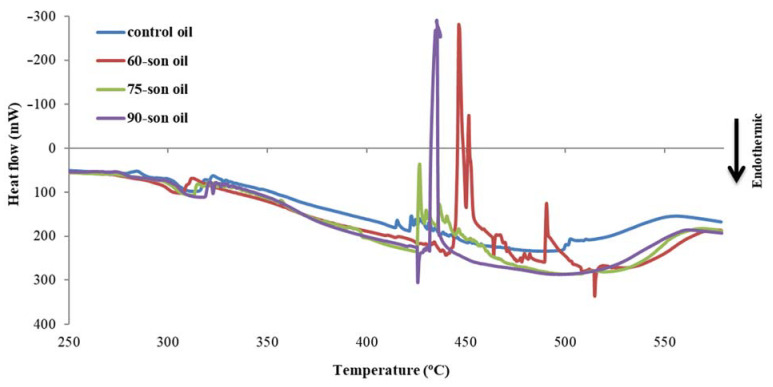
DSC diagram of enrichment sample oils; Curve of heat flow versus temperature.

**Figure 6 foods-12-00497-f006:**
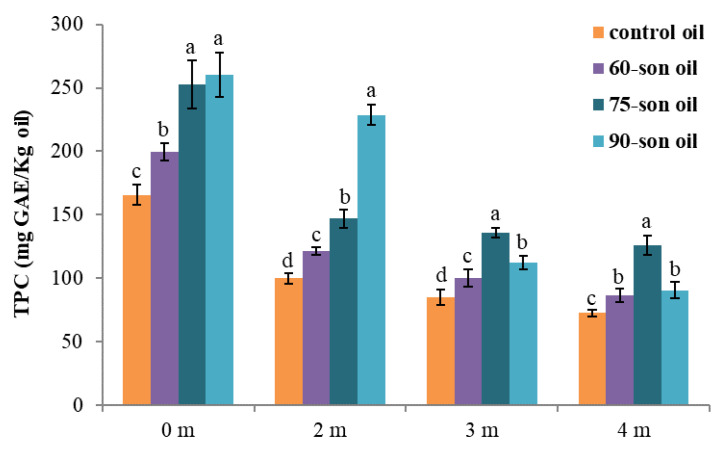
Total polyphenols in samples of 60-, 75-, and 90-son olive oils; Standard deviations are presented with error bars and means within each month with different superscript letters (e.g., a–d) are significantly (*p* < 0.05) different.

**Figure 7 foods-12-00497-f007:**
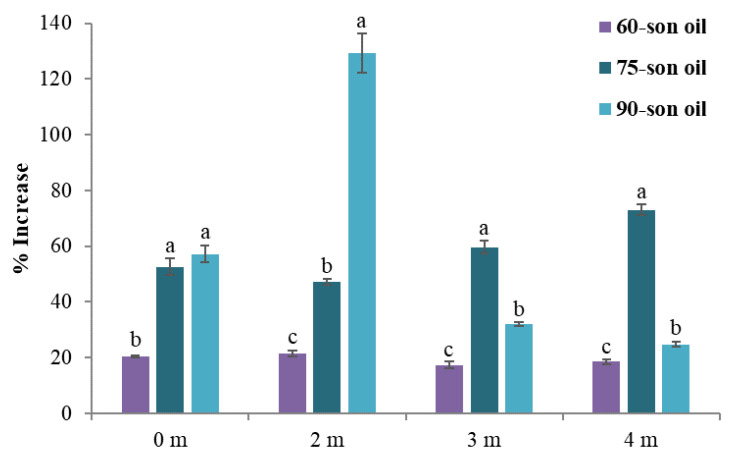
Increase (%) in total polyphenols after the enrichment; Standard deviation is presented with error bars and means within each month with different superscript letters (e.g., a–c) are significantly (*p* < 0.05) different.

**Table 1 foods-12-00497-t001:** Radical-scavenging effect (IC_50_) of micellar dispersions.

Samples	IC_50_
60-son	938.9 ± 33.8 ^a,^*
75-son	945.6 ± 45.4 ^a^
90-son	991.3 ± 69.4 ^a^
Total SP	1008.8 ± 25.2 ^a^

* Values are expressed as the mean values (±SD) of triplicate determinations and means within each column with different superscript letters (e.g., a) are significantly different (*p* < 0.05).

**Table 2 foods-12-00497-t002:** Physicochemical parameters of enriched olive oils.

Samples	Acidity	Ref. Index	Colorimetry			Conjugated Diene and Triene			Rancimat		DSC
	FFA (%)	nD20	*L**	*a**	*b**	K_232_	K_270_	ΔΚ	IT (h)	PF	*T*_max_ (°C)
control oil	0.790 ± 0.028 ^a,^*	1.4684 ± 0.0009 ^a^	71.4 ± 0.5 ^b^	−6.7 ± 0.3 ^b^	67.1 ± 0.4 ^a^	3.94 ± 0.25 ^a^	0.59 ± 0.04 ^b^	0.042 ± 0.002 ^a^	34.5 ± 2.1 ^a^	-	423 ± 27 ^a^
60-son oil	0.790 ± 0.017 ^a^	1.4683 ± 0.0006 ^a^	71.4 ± 0.4 ^b^	−5.9 ± 0.2 ^a^	66.3 ± 0.4 ^b^	3.78 ± 0.19 ^a^	0.66 ± 0.03 ^a,b^	0.021 ± 0.001 ^b^	33 ± 1.6 ^a^	0.96 ± 0.05 ^a^	455 ± 8 ^a^
75-son oil	0.733 ± 0.018 ^b^	1.4684 ± 0.0009 ^a^	72.5 ± 0.2 ^a^	−5.9 ± 0.2^a^	60.3 ± 0.1 ^c^	3.70 ± 0.14 ^a^	0.68 ± 0.05 ^a^	0.043 ± 0.003 ^a^	29.7 ± 1.7 ^b^	0.86 ± 0.03 ^b^	425 ± 10 ^a^
90-son oil	0.675 ± 0.022 ^c^	1.4682 ± 0.0010 ^a^	72.9 ± 0.5 ^a^	−5.9 ± 0.2^a^	60.0 ± 0.2 ^c^	3.89 ± 0.13 ^a^	0.64 ± 0.04 ^a,b^	0.044 ± 0.003 ^a^	29.8 ± 0.7 ^b^	0.86 ± 0.02 ^b^	435 ± 18 ^a^

* Values are expressed as the mean values (±SD) of triplicate determinations and means within each column with different superscript letters (e.g., a–c) are significantly (*p* < 0.05) different.

**Table 3 foods-12-00497-t003:** Radical-scavenging effect (IC_50_) of micellar dispersion-enhanced oil samples.

Samples	IC_50_
control oil	205,400 ± 12,529 ^b,^*
60-son oil	219,050 ± 8,105 ^b^
75-son oil	154,650 ± 11,599 ^c^
90-son oil	359,000 ± 20,104 ^a^

* Values are expressed as the mean values (±SD) of triplicate determinations and means within each column with different superscript letters (e.g., a–c) are significantly different (*p* < 0.05).

## Data Availability

All the data are contained within the article.

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
