# Peer review of "Development of Enriched Oil with Polyphenols Extracted from Olive Mill Wastewater"

_foods, 2023, doi:10.3390/foods12030497_

Round 1

Reviewer 1 Report

The presented paper contains an interesting hypothesis worth further continuation in the form of further research. The topic is very relevant. The paper presents novel and useful findings, is sound and is well structured, and follows a logical sequence. The introduction provides evidence-based background for the research. The methodology consists of straight-forward methodologies, and deals with measurements of key parameters. The methods have been properly described. The literature is well represented, and the discussion is sufficiently elaborated for that kind of topic, with a clear (positive) conclusion. The results are well presented and data interpretation is appropriate. Results were properly reported and the findings have been accurately discussed and compared with other published papers. The findings are thoroughly discussed, and conclusions are justified by the results. The manuscript was properly conducted and findings reported are important. The paper contains valuable data. The authors investigated an interesting topic and the objective of the paper is of worldwide interest and fits well within the overall scope of the journal. I did not find any objective errors.

Author Response

We would like to thank the reviewer for his/her comments.

Reviewer 2 Report

I reviewed the manuscript entitled, Development of Enriched Oil with Polyphenols Extracted from Olive Mill Wastewater. Although extraction of polyphenolic compounds from olive mill waste water Is not a novel concept, authors used polyphenols to enrich the oil's efficacy, where I can see little contribution.

Lines 23-28: please separate research objectives and findings. There is no clarity as such. Also, provide an industrial application and contribution to the field.

Introduction is appropriate and clearly explains the need of conducting this study

Materials and methods

Section 2.3. Isolation can be revised to Extraction

Figure 1: isolation of polyphenols….. can be revised as extraction of polyphenols

References are not according to the journal format. Extensive editing is required.

Section 2.5 should be section 2.4 CPE method performance determination and 2.5 should be Enrichment of olive oil with micellar dispersions

Results and discussion

Line 234: What is the Total SP diagram?

Place below lines after the figure caption

Line 284: In Fig. 4 shows the enriched olive oil.. very small sentence

Table 1. Radical-scavenging effect (IC50) of micellar dispersions… why all micellar showed no significant effect, which means there is no effect of 60-son and others ?

The discussion is appropriate with available literature and compared

Conclusions should be revised to reflect the study findings

References should be cross-checked. Some of the references are not in accordance with journal guidelines. All scientific names must be in italics 

Author Response

I reviewed the manuscript entitled, Development of Enriched Oil with Polyphenols Extracted from Olive Mill Wastewater. Although extraction of polyphenolic compounds from olive mill waste water Is not a novel concept, authors used polyphenols to enrich the oil's efficacy, where I can see little contribution.

-We would like to thank the reviewer for his/her comments.

Lines 23-28: please separate research objectives and findings. There is no clarity as such. Also, provide an industrial application and contribution to the field.

-The abstract has been revised, as suggested.

Introduction is appropriate and clearly explains the need of conducting this study

-Thank you very much.

Materials and methods

Section 2.3. Isolation can be revised to Extraction

-Isolation was revised to Extraction, as suggested.

Figure 1: isolation of polyphenols….. can be revised as extraction of polyphenols

-Isolation was revised to Extraction, as suggested.

References are not according to the journal format. Extensive editing is required.

-References were checked and revised according to the journal format.

Section 2.5 should be section 2.4 CPE method performance determination and 2.5 should be Enrichment of olive oil with micellar dispersions

-Thank you for the correction. The sections have changed.

Results and discussion

Line 234: What is the Total SP diagram?

-Diagram was revised to a column. The Total SP column represents the total micellar phase of the three recoveries.

Place below lines after the figure caption

-Corrected.

Line 284: In Fig. 4 shows the enriched olive oil.. very small sentence

-The sentence has changed. The new sentence in L265 is ‘’The enriched olive oil is shown in Fig. 4, which also shows how the organoleptic characteristics of the olive oil changed’’.

Table 1. Radical-scavenging effect (IC50) of micellar dispersions… why all micellar showed no significant effect, which means there is no effect of 60-son and others ?

-The micellar dispersions showed no statistically significant differences (p < 0.05), indicating that the sonication duration does not affect the radical-scavenging effect. The following sentence was added to L275: ‘’Statistical analysis, however, showed that there was no significant difference (p < 0.05) between the three sonication duration levels’’.

The discussion is appropriate with available literature and compared

-Thank you.

Conclusions should be revised to reflect the study findings

-The conclusions have been revised.

References should be cross-checked. Some of the references are not in accordance with journal guidelines. All scientific names must be in italics 

-References were cross-checked and revised according to the journal format.

Reviewer 3 Report

This study is about recovery of phenolic compounds from olive mill waste and its uses in the fortification of olive oils. It is interesting subject and can have good impact on the environment and also in the nutritional value of the oils.

It could be suggested to use other oil rather than olive oil such as canol, sunflower etc to fortification as olive oil has its own phenolic compounds and generally, phenolic compounds in the waste are hydrophilic and have limited solubility. Also, this should be noted that phenolic compounds at higher concentration may act as pro-oxidant instead of being antioxidant.

In Abstract section, from line 15 to 23 is the description and presentation of the study. It could be shorter to write more about the methods, data and results section.

Line 81-82: Enriching olive oil with polyphenols is intended to increase its concentration without increasing calories. There is no need to write this statement as there is no place to discuss about calories.

Materials section, line 91, the type of olive oil extraction method should be written.  Line 130: Type of olive oil used should be identified as there are man types of olive oils. The olive oil composition should be clearly presented. Line 131: the reason behind using 0.5, 1 and then increasing level to 5 % without using 2 or 3 % should be clarified. 

Results and discussion section need serious and extensive revision, it is written in a complex way and should be rewritten. It was so difficult to go through this section and give comments as it has complexity line by line.

Author Response

This study is about recovery of phenolic compounds from olive mill waste and its uses in the fortification of olive oils. It is interesting subject and can have good impact on the environment and also in the nutritional value of the oils.

-We would like to thank the reviewer for his/her comments.

It could be suggested to use other oil rather than olive oil such as canol, sunflower etc to fortification as olive oil has its own phenolic compounds and generally, phenolic compounds in the waste are hydrophilic and have limited solubility. Also, this should be noted that phenolic compounds at higher concentration may act as pro-oxidant instead of being antioxidant.

-Thank you for your suggestion. We will keep this in mind for future studies. However, the olive oil used was a mixture of is virgin and refined olive oils, not an EVOO. Its total polyphenols were only 60 mg GAE/Kg. Also, thanks for the valuable comment. The following sentence was added to L82: ‘’However, the antioxidant/pro-oxidant balance depends on many factors, particularly dose (e.g., concentration) and timing (e.g., storage duration) and it is a very delicate process that can readily be changed (Chedea et al., 2021)’’.

  • Chedea, V. S., TomoiagÇŽ, L. L., Macovei, Åž. O., MÇŽgureanu, D. C., Iliescu, M. L., Bocsan, I. C., ... & Pop, R. M. (2021). Antioxidant/pro-oxidant Actions of Polyphenols from grapevine and Wine By-Products-Base for Complementary Therapy in Ischemic Heart Diseases. Frontiers in Cardiovascular Medicine, 8, 750508.

In Abstract section, from line 15 to 23 is the description and presentation of the study. It could be shorter to write more about the methods, data and results section.

-The abstract has been revised, as suggested.

Line 81-82: Enriching olive oil with polyphenols is intended to increase its concentration without increasing calories. There is no need to write this statement as there is no place to discuss about calories.

-The statement was removed, as suggested.

Materials section, line 91, the type of olive oil extraction method should be written.  Line 130: Type of olive oil used should be identified as there are man types of olive oils. The olive oil composition should be clearly presented. Line 131: the reason behind using 0.5, 1 and then increasing level to 5 % without using 2 or 3 % should be clarified. 

-The type of olive oil was written in L96, in the first edition. The olive oil used is composed of virgin and refined olive oils. Following that, this olive oil was analyzed as a control sample with the enriched samples under many quality methods of analysis. Also, regarding the selection of these enrichment levels, we wanted a higher concentration to demonstrate that the solubilization of phenolic compounds on surfactant micelles becomes saturated at higher surfactant concentrations, and the polyphenols retention decreases. This finding is according to the study by Víctor-Ortega et al. (2017) and presented in L228 in our manuscript.

  • Víctor-Ortega, M. D., Martins, R. C., Gando-Ferreira, L. M., & Quinta-Ferreira, R. M. (2017). Recovery of phenolic compounds from wastewaters through micellar enhanced ultrafiltration. Colloids and Surfaces A: Physicochemical and Engineering Aspects, 531, 18-24.

Results and discussion section need serious and extensive revision, it is written in a complex way and should be rewritten. It was so difficult to go through this section and give comments as it has complexity line by line.

-We are truly sorry for this confusion. The results and discussion section have been improved in quality.

Round 2

Reviewer 3 Report

The comments and suggestions are included in the revised version of the manuscript and its acceptance is suggested.